# Cannabinoids for the Treatment of Hair, Scalp, and Skin Disorders: A Systematic Review

**Meagan Popp** [1] , **Steven Latta** [1] , **Betty Nguyen** [2,]*, **Colombina Vincenzi** [3] **and Antonella Tosti** [2]

1    Herbert Wertheim College of Medicine, Miami, FL 33199, USA
2    Dr. Phillip Frost Department of Dermatology and Cutaneous Surgery, University of Miami Miller School of Medicine, Miami, FL 33136, USA
3    Dermatology Unit, IRCCS Azienda Ospedaliero Universitaria di Bologna, 40138 Bologna, Italy
*    Correspondence: bettynguyenmed@gmail.com

**Abstract:** Cannabinoid products have been studied in the treatment of various dermatologic conditions. We searched PubMed/MEDLINE for articles published before 1 February 2023 that described the use of cannabinoids in the management of hair, scalp, and skin conditions, identifying 18 original articles that encompassed 1090 patients who used various forms of cannabinoid products. Where specified, topical cannabidiol (CBD) was the most commonly utilized treatment (64.3%, 173/269), followed by oral dronabinol (14.4%, 39/269), oral lenabasum (14.1%, 38/269), and oral hempseed oil (5.9%, 16/269). Using the GRADE approach, we found moderate-quality evidence supporting the efficacy of cannabinoid products in managing atopic dermatitis, dermatomyositis, psoriasis, and systemic sclerosis and moderate-quality evidence supporting a lack of efficacy in treating trichotillomania. There was low to very low quality evidence supporting the efficacy of cannabinoid products in managing alopecia areata, epidermolysis bullosa, hyperhidrosis, seborrheic dermatitis, and pruritus. Our findings suggest that cannabinoids may have efficacy in managing symptoms of certain inflammatory dermatologic conditions. However, the evidence is still limited, and there is no standardized dosage or route of administration for these products. Large randomized controlled trials and further studies with standardized treatment regimens are necessary to better understand the safety and efficacy of cannabinoids.

**Keywords:** cannabinoid; cannabis; CBD; hair; scalp; skin; therapy; treatment; alopecia areata; atopic dermatitis; dermatomyositis; epidermolysis bullosa; hyperhidrosis; psoriasis; seborrheic dermatitis; systemic sclerosis; trichotillomania

## 1. Introduction

Cannabinoid products, which contain substances from the cannabis plant, have become increasingly accessible on the consumer market as a potential treatment for various conditions, and their use has been studied in the management of various skin- and hair-related symptoms [1–3]. Cannabinoids can be found endogenously (endocannabinoids), derived from plants (phytocannabinoids), or made synthetically. There are over 100 different cannabinoids found in the cannabis plant, but the most well-known and commonly available types of phytocannabinoids for consumers include delta-9-tetrahydrocannabinol (THC), cannabidiol (CBD), and cannabinol (CBN) [4,5]. Common forms of synthetic cannabinoids include dronabinol (a synthetic THC) and lenabasum (a synthetic cannabinoid analog).

Cannabinoid receptors are G-protein coupled receptors that play an important role in skin homeostasis through the endocannabinoid system (ECS), which comprises a network of receptors, neurotransmitters, and enzymes that help to regulate several functions in the body [4]. There are two main types of cannabinoid receptors in the ECS: cannabinoid receptor type 1 (CB1) receptors, which are primarily expressed in the brain and peripheral

nervous system; and cannabinoid receptor type 2 (CB2) receptors, which are primarily expressed in activated immune cells and peripheral tissues and are thought to be a regulator of inflammation [6]. CB2 receptors are also expressed in epidermal keratinocytes, dermal cells, cutaneous nerve fibers, melanocytes, eccrine sweat glands, and hair follicles, and the activation of CB2 receptors has been shown to lead to decreased proliferation, pain, itch, and inflammation [4,7–9]. Unlike THC, which exerts psychoactive effects through its partial agonist activity against CB1 receptors, CBD has little binding affinity to CB1 or CB2 receptors and therefore does not exert psychoactive effects [6,9,10].

Cannabinoids have been used in the management of various inflammatory conditions, as they have been shown to inhibit the release of pro-inflammatory cytokines, including interleukin (IL)-6, IL-8, and tumor necrosis factor (TNF)-alpha [11]. In patients with multiple sclerosis, cannabinoids have been reported to decrease spasticity, pain, and bladder dysfunction compared to placebo [12]. In patients with inflammatory bowel disease, cannabinoid use was reported to decrease the Crohn's Disease Activity Index and improve patients' quality of life compared to a placebo [13]. Within dermatology, cannabinoids have been studied as a treatment for inflammatory conditions including atopic dermatitis, epidermolysis bullosa, dermatomyositis, psoriasis, pruritus, scalp psoriasis, and systemic sclerosis. In an in vitro model, cannabinoids including CBD, CBN, and THC were shown to suppress human keratinocyte proliferation in a concentration-dependent manner [14].

As of February 2022, 37 states, Washington, D.C., and 3 territories of the United States, as well as countries including the Netherlands, Germany, and Italy, have legalized the use of medical cannabis [15]. The U.S. Food and Drug Administration (FDA) has approved several medications containing cannabinoids for the treatment of seizures, loss of appetite in people with HIV/AIDS, and nausea and vomiting secondary to chemotherapy [16], but treatments have not yet been approved for dermatologic conditions.

Various reports have described cannabinoid products in the treatment of dermatologic conditions, but few have systematically evaluated their scope and efficacy. We have systematically reviewed the literature to characterize the potential applications of cannabinoid therapy for the treatment of dermatologic conditions.

## 2. Methods

Study identification was conducted according to the Preferred Reporting Items for Systematic Reviews and Meta-Analyses (PRISMA) guidelines (Figure 1). A comprehensive literature search was conducted on PubMed/MEDLINE for articles published before 1 February 2023. Key search terms included "cannabis", "canabis", "cannabidiol", "marijuana", "marihuana", "tetrahydrocannabinol", AND "hair", "scalp", "trichotillomania", "skin", "itch", "pruritus", "dermatology", AND "treatment", "therapy", NOT "animal". Two reviewers (M.P. and S.L.) independently screened articles using titles and abstracts to remove animal studies and duplicate, abstract-only, non-English, and review articles, yielding 530 articles. Articles were further excluded if no full text was available or if they lacked direct relevance to cannabinoids in the treatment of hair, scalp, or skin disease. A third reviewer (B.N.) resolved any discrepancies.

The GRADE approach (grading of recommendations, assessment, development, and evaluations) was used to grade the quality of evidence of studies included in this review. The GRADE approach is a systematic approach that categorizes a study's quality of evidence as: (1) very low, (2) low, (3) moderate, or (4) high. Randomized controlled trials (RCTs) start as high-quality evidence, whereas observational studies start as low-quality evidence. Five factors can decrease the quality of evidence (risk of bias, inconsistency, indirectness, imprecision, and publication bias), and three factors can increase the quality of evidence (large effect, dose response, and all plausible confounding variables considered).

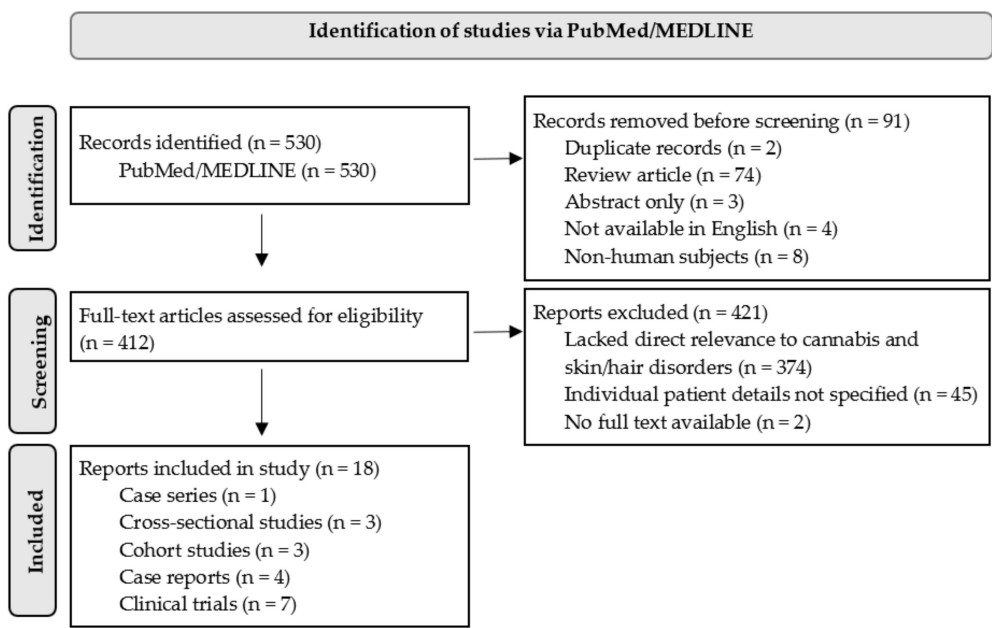

**Figure 1.** Flowchart of study identification via PubMed/MEDLINE according to the Preferred Reporting Items for Systematic Reviews and Meta-Analyses (PRISMA) guidelines.

## 3. Results

A total of 18 reports (7 clinical trials, 4 case reports, 3 cohort studies, 3 cross-sectional studies, and 1 case series) were included in this review. Table 1 includes the descriptions of each study including the type of dermatologic condition investigated, treatment, and results. Using the GRADE approach, we found moderate-quality evidence supporting the efficacy of cannabinoid products in atopic dermatitis [17–19], dermatomyositis [5], psoriasis [20–23], seborrheic dermatitis [22], and systemic sclerosis [24] and moderate quality evidence supporting a lack of efficacy in trichotillomania [25,26]. There was low to very low quality evidence supporting the efficacy of cannabinoid products in alopecia areata [27], epidermolysis bullosa [28,29], hyperhidrosis [30], and pruritus [31–33] (Table 1).

**Table 1.** Studies on cannabinoid-based products for the treatment of hair, scalp, and skin conditions.

| Condition | Author; Year | Type of Study, Number of Participants, Age (Years), (Sex, Male:Female) | Treatment | Duration of Treatment | Response to Treatment | GRADE Rating |
|---|---|---|---|---|---|---|
| AA | Han; 2022 [27] | Cross-sectional study, *n* = 1045 Mean age: 47.6 (172 M:870 F: 3 declined to specify) | 689/1045 endorsed use of cannabinoid products, including smoking marijuana or CBD, ingesting marijuana, THC or CBD, inhaling vaporized liquid THC, hash oil, or CBD, and CBD lotions and creams | Varies | 80.4% (*n* = 287) no change in hair loss, 37.8% (*n* = 135) no change in discomfort of skin | Very low |

**Table 1.** *Cont.*

| Condition | Author; Year | Type of Study, Number of Participants, Age (Years), (Sex, Male:Female) | Treatment | Duration of Treatment | Response to Treatment | GRADE Rating |
|---|---|---|---|---|---|---|
| AD | Gao; 2022 [17] | Randomized clinical trial, *n* = 57 Age range: 18–65 (not provided) | Randomized to Group 1: JW-100 (Jupiter Wellness, Inc.) (*n* = 18), topical CBD from hemp with aspartame; Group 2: pure topical CBD from hemp (*n* = 17); or Group 3: placebo (*n* = 17), in a 1:1:1 ratio. | Twice daily for 14 days | Efficacy was scored using the Investigator's Static Global Assessment (ISGA), which grades disease severity based on morphologic appearances on a scale from 0 to 4. JW-100 group demonstrated the most significant reduction in ISGA (1.28, *p* = 0.042) versus placebo.<br><br>50% of patients in the JW-100 group achieved clear or almost clear scores compared with 15% in the placebo group (*p* = 0.028)<br><br>No statistically significant improvement in Group 2 compared to placebo (*p* = 0.727). | Moderate |
| | Maghfour; 2021 [18] | Cohort study, *n* = 14 Mean age: 51.36 (11 M:3 F) | Topical 1% CBD gel and hemp oil containing 1% CBD | 14 days | Reduction in mean EASI score (5.40 at baseline and 3.10 at end of two weeks) (*p* < 0.005) | Low |
| | Callaway; 2005 [19] | Crossover randomized clinical trial, *n* = 16 Age range: 25–65 (1 M:15 F) | Oral consumption of 30 mL hempseed oil or olive oil daily | 8 weeks of each treatment with 4-week washout and crossover period | TEWL values decreased from baseline in the hempseed oil group at 8 weeks (*p* = 0.074), but there was no statistically significant difference between hempseed and olive oil groups at 8 weeks (*p* = 0.274). Patient reported skin dryness and itchiness improved (*p* = 0.027 and *p* = 0.023, respectively) after hempseed oil intervention | Moderate |

Table 1. *Cont.*

| Condition | Author; Year | Type of Study, Number of Participants, Age (Years), (Sex, Male:Female) | Treatment | Duration of Treatment | Response to Treatment | GRADE Rating |
|---|---|---|---|---|---|---|
| DM | Werth; 2022 [5] | Randomized clinical trial, *n* = 22 (11 received lenabasum, 11 received placebo) Mean age of lenabasum group: 53.1 (12 M:10 F) | Oral lenabasum 20 mg daily for 28 days and then 20 mg twice per day for 56 days or placebo | 113 days | On Day 113, the adjusted mean (SD) change from baseline CDASI activity score was −9.3 (10.99) in the lenabasum group and −3.7 (6.83) in the placebo (*p* = 0.0382). Treatment with lenabasum resulted in statistically significant reductions from the baseline in IFN-b and IFN-g levels (*p* = 0.030 and *p* = 0.048, respectively) | Moderate |
| EB | Schäder; 2021 [28] | Cross-sectional study, *n* = 71 Age: Not reported (40 M:31 F) | Topical, ingested, inhaled, and sublingual cannabinoid-based medicines containing CBD only (*n* = 24/118), THC only (*n* = 18/118), THC/CBD (*n* = 41/118), and unspecified cannabinoids | Variable (<6 months; >5 years) | Statistically significant reductions in self-reported pain and pruritus (median pain change-score: − 3, *p* < 0.001; median pruritus change-score: − 3, *p* < 0.001). Patient-reported improvement in overall EB symptoms (95.8%, 46/48), pain (93.8%, 45/48), pruritus (90.9%, 40/44), skin inflammation (72.3%, 34/47) and wound-healing time (60.4%, 29/48) | Low |
| | Chelliah; 2018 [29] | Case series, *n* = 3, Age range: 6 months to 10 years (2 M:1 F) | Topical CBD oil | Varies | Reported reduction in pain and blistering in 3 patients | Low |
| Hyperhidrosis | Kaemmerer; 2022 [30] | Case report, *n* = 1 Age: 28 (1 M) | Topical dronabinol drops 25 mg/mL up to three times daily for one month, inhaled 0.5 g medical cannabis buds (Pedanios 8% THC and 8% CBD) for two weeks, and vaporized 0.5 g medical cannabis buds (Pedanios 20% THC, 1% CBD) for two weeks | 2 months | 55.6% decrease in DLQI score (10-point decrease); 25% increase in EQ-5D-3L score (2-point improvement); 40% increase in EQ VAS score (20-point improvement). HDSS decreased by 80% (2-point decrease). | Very low |

**Table 1.** *Cont.*

| Condition | Author; Year | Type of Study, Number of Participants, Age (Years), (Sex, Male:Female) | Treatment | Duration of Treatment | Response to Treatment | GRADE Rating |
|---|---|---|---|---|---|---|
| Pruritus | Roh; 2021 [31] | Case report, n = 1 Age: "60s" (1 F) | Smoking THC or indica flower and sublingual indica flower or tincture form (THC and cannabinol compounded in 1:1 ratio) two nights weekly | 20 months | DLQI score reduction from 17 at baseline to 1 at 20 months. | Very low |
|  | Lou; 2021 [32] | Case report, n = 1 Age: 60 (1 M) | Oral capsule of 2.43 mg THC/CBD 2.75 mg once to twice daily | 2 weeks | Pruritus score decreased from 7/10 to 3/10 | Very Low |
|  | Mahurin; 2022 [33] | Cross-sectional study, n = 119 Mean age: 59 (39 M:65 F: 2 decline to specify) | 60 participants endorsed use of non-specific cannabinoid products (smoking, vaporizing, topical creams/ointments, and oral) | Variable | 25% (6/24) of current users reported using cannabis specifically to treat itch. These users reported moderate improvement in itch (VAS scores for degree of symptom improvement mean of 6.6/10) | Low |
| Psoriasis | Puaratanaarunkon; 2022 [20] | Split-body randomized controlled trial, n = 51 (108 pairs of target plaques) Mean age: 53 (30 M:21 F) | Topical 2.5% CBD ointment or placebo twice daily on target plaques | 12 weeks | Significantly lower difference in PASI score ($p = 0.026$); 10% higher grade reduction than placebo | Moderate |
|  | Friedman; 2020 [21] | Case report, n = 1 Age: 33 (1 M) | THC distillate cream with medium-chain triglyceride oil, THC soap, hair oil with THC distillate dissolved into jojoba oil, 5 mg/mL daily | Continuous use for 14 days; every few days thereafter for maintenance | Patient reported resolution of symptoms after two months | Very low |
| Scalp psoriasis or SD | Vincenzi; 2020 [22] | Cohort study, n = 50 Mean age: 42.16 (24 M:26 F) | Topical shampoo containing 150 mg CBD/205 mL | 14 days | Severity scores of arborizing vessels, twisted capillaries, and scales reduced from 2.3 to 0.5, 2.6 to 0.8, and 3.6 to 0.6, respectively (all $p < 0.0001$). Severity scores for erythema and scaling reduced from 5.5 to 1.3 and 7.0 to 1.6, respectively (both $p < 0.0001$). | Low |
| Psoriasis, AD, and resulting scars | Palmieri; 2019 [23] | Cohort study, n = 20 Age range: 20–80 (6 M:14 F) | Topical CBD-enriched ointment (hemptouch organic skin care) twice daily | 3 months | Improvement in PASI score ($p < 0.001$). Hydration increased ($p < 0.01$), elasticity improved ($p < 0.001$), and TEWL improved ($p < 0.001$) | Low |

**Table 1.** *Cont.*

| Condition | Author; Year | Type of Study, Number of Participants, Age (Years), (Sex, Male:Female) | Treatment | Duration of Treatment | Response to Treatment | GRADE Rating |
|---|---|---|---|---|---|---|
| SSc | Spiera; 2020 [24] | Randomized clinical trial, $n = 42$ (27 received lenabasum, 15 received placebo) Mean age of lenabasum group: 49 (10 M:32 F) | Oral lenabasum 5 mg once daily, 20 mg once daily, or 20 mg twice daily for 4 weeks, followed by 20 mg twice daily for 8 weeks vs. placebo (microcrystalline cellulose) | 16 weeks | Median CRISS score significantly improved in lenabasum group compared to placebo at Week 16 ($p = 0.04$ by one-sided MMRM analysis and $p = 0.07$ by two-sided MMRM analysis) | Moderate |
| TTM | Grant; 2011 [25] | Open-label clinical trial, $n = 14$ Mean age: 33.3 (0 M:14 F) | All patients were started on oral dronabinol (dose ranging from 2.5 to 15 mg/day). No control group | 12 weeks | MGH-HPS scores decreased from a mean of $16.5 \pm 4.4$ at baseline to $8.7 \pm 5.5$ at Week 12 ($p = 0.001$) | Low |
| TTM and skin-picking disorder | Grant; 2022 [26] | Randomized clinical trial, $n = 50$ (trichotillomania $n = 34$; skin picking disorder $n = 16$) Age: 33.04 (6 M:19 F) | Oral dronabinol (5–15 mg/day) ($n = 25$) vs. placebo ($n = 25$) | 10 weeks | No statistically significant change in outcomes, as measured by the clinician-rated National Institute of Mental Health scale for hair pulling or skin picking | Moderate |

AA, alopecia areata; AD, atopic dermatitis; CBD, cannabidiol; CDASI, Cutaneous Dermatomyositis Disease Area and Severity Index; CRISS, Combined Response Index in Diffuse Cutaneous Systemic Sclerosis; DLQI, Dermatology Life Quality Index; DM, dermatomyositis; EB, epidermolysis bullosa; EQ-5D-3L, European Quality of Life 5 Dimensions 3 Level Version; HDSS, Hyperhidrosis Disease Severity Scale; MGH-HPS, Massachusetts General Hospital Hair Pulling Scale; MMRM, mixed-effects model repeated measures; PASI, Psoriasis Area Severity Index; SD, seborrheic dermatitis; SSc, systemic sclerosis; TEWL, transepidermal water loss; THC, tetrahydrocannabinol; TTM, trichotillomania; VAS, Visual Analog Scale.

Types of cannabinoids used. Of the 1578 patients analyzed in this review, 1090 used various forms and administrations of cannabinoid products (Table 2). Where specified, topical CBD was the most commonly utilized treatment (64.3%, 173/269), followed by oral dronabinol (14.4%, 39/269), oral lenabasum (14.1%, 38/269), and oral hempseed oil (5.9%, 16/269) (Table 2).

Routes of administration. Of the 18 reports identified, 7 studies (3 cohort studies, 2 clinical trials, and 2 case series/reports) evaluated *topical* cannabinoids, 6 studies (5 clinical trials, 1 case report) evaluated *oral* cannabinoids, 1 case report evaluated *smoking* cannabinoids, and 4 studies (3 cross-sectional, 1 case report) evaluated a *mix* of topical, oral, inhaled, or vaporized cannabinoids. Where specified, 64.7% (174/269) of patients used topical cannabinoids, 34.9% (94/269) of patients used oral cannabinoids, and 0.4% (1/269) of patients smoked cannabinoids (Table 2).

Safety. Topical cannabinoids were safe to use, and the most commonly reported side effect of topical cannabinoids was mild site irritation [17,18,20]. The side effects reported with the oral administration of cannabinoids, specifically oral lenabasum, included mild dizziness, fatigue, dry mouth, and diarrhea [5,24].

**Table 2.** Number of patients who used various types of cannabinoid treatments and their routes of administration.

| Types of Cannabinoid Used | Route of Administration | Number of Patients | Conditions Investigated (Number of Patients Treated) |
|---|---|---|---|
| CBD only | Topical | 173 | Atopic dermatitis ($n = 49$) [17,18], Epidermolysis bullosa ($n = 3$) [29], Psoriasis ($n = 71$) [20,23], Scalp psoriasis and seborrheic dermatitis ($n = 50$) [22] |
| Dronabinol | Oral | 39 | Trichotillomania ($n = 39$) [25,26] |
| Hempseed oil | Oral | 16 | Atopic dermatitis ($n = 16$) [19] |
| Lenabasum | Oral | 38 | Dermatomyositis ($n = 11$) [5], Diffuse cutaneous systemic sclerosis ($n = 27$) [24] |
| THC and *Cannabis indica* flower | Smoking | 1 | Pruritus ($n = 1$) [31] |
| THC only | Topical | 1 | Psoriasis ($n = 1$) [21] |
| THC/CBD combination | Oral | 1 | Pruritus ($n = 1$) [32] |
| Non-specific | Topical, ingested, inhaled, sublingual | 821 | Alopecia areata ($n = 689$) [27], Epidermolysis bullosa ($n = 71$) [28], Hyperhidrosis ($n = 1$) [30], Pruritus secondary to cutaneous lymphoma ($n = 60$) [33] |
| Total | | 1090 | |

CBD, cannabidiol; THC, tetrahydrocannabinol.

## 4. Discussion

Atopic dermatitis. We identified one clinical trial and two cohort studies that evaluated the efficacy of topical CBD with and without aspartame, topical hemp oil, and topical hempseed oil in treating atopic dermatitis. In these three studies, while topical CBD itself did not yield statistically significant results, CBD with aspartame (ASP) [17], 1% CBD with hemp oil [18], and hempseed oil alone [19] (which typically contains no more than trace amounts of CBD) led to statistically significant clinical improvements. In a double-blind randomized clinical trial of 57 patients, the efficacy of combined topical CBD and ASP twice daily for 14 days was compared to a placebo. Of participants who received the CBD and ASP combination treatment, 50% experienced clear-to-almost-clear Investigator's Static Global Assessment (ISGA) scores, compared to 15% who were administered a placebo ($p = 0.042$) [17]. However, the group using topical CBD alone did not have a statistically significant reduction in ISGA score compared to the placebo ($p = 0.727$) [17]. In a cohort study of 14 patients treated with 1% CBD-infused gel and hemp oil containing 1% CBD, a statistically significant reduction in the mean Eczema Area Severity Index (EASI) score, from 5.4 at baseline to 3.10 (out of a maximum score of 12), was found at the end of the two-week trial ($p < 0.005$) [18]. Five participants (35%) reported discomfort and stinging upon the application of topical CBD. In the final cohort study, 16 participants were treated with hempseed oil for 20 weeks and had decreased transepidermal water loss (TEWL) and skin itchiness after 20 weeks of treatment ($p = 0.074$) [19].

Dermatomyositis. We identified one randomized clinical trial that evaluated the efficacy of oral lenabasum in treating dermatomyositis. In this study, 22 participants were given either oral lenabasum (20 mg once daily for 28 days then twice daily on Days 29–84) or a placebo [5]. A change in the baseline Cutaneous Dermatomyositis Disease Area and Severity Index (CDASI) score was the primary endpoint. At the end of the study (Day 113, four weeks after stopping oral lenabasum), 18.2% (2/11) had CDASI scores <14 and 27.3% (3/11) had CDASI scores between 14 and 19, whereas all participants in the placebo group had scores ≥20 ($p = 0.0351$, two-sided exact test). The mean (SD) change from the baseline CDASI score was −9.3 (10.99) in the lenabasum group and −3.7 (6.83) in the placebo ($p = 0.0382$), suggesting that lenabasum is associated with improvement in the symptoms

of dermatomyositis. The side effects reported included mild dizziness, fatigue, dry mouth, and diarrhea. One patient reported abnormal dreams and another reported depressed mood, irritability, and agitation, though no patients reported psychoactive adverse events.

Psoriasis. We identified one clinical trial, two cohort studies, and one case report that evaluated the efficacy of various cannabinoid products in treating psoriasis and scalp psoriasis. In a double-blind, randomized clinical trial, 51 patients (108 target plaques) were treated with 2.5% CBD ointment on one side of the body versus placebo on the other for 12 weeks [20]. The Psoriasis Area Severity Index (PASI) score, graded on a scale of 0–4 based on erythema, scaling/desquamation, and induration, was measured. By the 12-week follow up, the mean PASI score on the side of the body treated with CBD was significantly lower than the placebo by 0.197 ($p = 0.026$). Six patients experienced skin irritation that resolved within one week after cessation. In a cohort study of participants with psoriasis ($n = 5$), atopic dermatitis ($n = 5$), or resulting scarring ($n = 10$), participants were treated with CBD-enriched ointment twice daily for 3 months [23]. All participants reported a significant improvement in skin hydration ($p < 0.01$) and in TEWL ($p < 0.001$) in the forehead, bilateral malar area, and bilateral neck. There was also an improvement in skin elasticity at four of the five aforementioned sites ($p < 0.001$). On Day 90, the PASI score significantly improved compared to baseline ($p < 0.001$).

Scalp psoriasis and seborrheic dermatitis. In a cohort study of 50 patients with mild to moderate scalp psoriasis ($n = 22$) and seborrheic dermatitis ($n = 28$) [22], an anti-inflammatory shampoo (Revita; CBD, DS Laboratories) containing broad spectrum CBD and ketoconazole was used for 14 days and resulted in a significant decrease in inflammation across several parameters. A six-point scale was used to rate the severity of arborizing vessels, twisted capillaries, and scales assessed by videodermoscopy. Ten-point scales were used to grade the severity of itching and burning and clinical signs of erythema and scaling. Following the two-week treatment, the severity scores of arborizing vessels, twisted capillaries, and scales reduced from 2.3 to 0.5, 2.6 to 0.8, and 3.6 to 0.6, respectively (all $p < 0.0001$). Severity scores for erythema and scaling decreased from 5.5 to 1.3 and 7.0 to 1.6, respectively (both $p < 0.0001$). Scores for itching and burning at Week 2 decreased from 6.9 to 1.5 and 4.5 to 1.0, respectively (both $p < 0.0001$). Given the various ingredients contained in this shampoo, further studies are needed to isolate the effects of CBD.

Systemic sclerosis. We identified one randomized clinical trial that evaluated the efficacy of oral lenabasum in managing systemic sclerosis [24]. Patients were treated with oral lenabasum at varying dosages ($n = 27$) or placebo ($n = 15$) for 8 weeks. The American College of Rheumatology Combined Response Index in Systemic Sclerosis (CRISS) score was used to measure efficacy. At Week 16, the median CRISS score improved in the lenabasum group (0.33) but not the placebo (0.00) ($p = 0.04$ by one-sided MMRM analysis and $p = 0.07$ by two-sided MMRM analysis). Adverse events that occurred included dizziness (22%), fatigue (19%), headache (11%), and arthralgia (11%) and these were mild except for one case of moderate dizziness.

## 5. Conclusions

Our findings suggest that cannabinoids may have efficacy in managing symptoms of inflammatory dermatologic conditions, including atopic dermatitis, dermatomyositis, psoriasis, and systemic sclerosis. However, evidence is still limited, and there is no standardized dosage or route of administration for these products [34]. Large randomized controlled trials and further studies with standardized treatment regimens are necessary to better understand the safety and efficacy of cannabinoids.

**Author Contributions:** Conceptualization: A.T. and B.N.; Data collection and analysis: M.P., S.L. and B.N.; Writing: M.P., S.L., B.N., C.V. and A.T. All authors have read and agreed to the published version of the manuscript.

**Funding:** This research received no external funding.

**Institutional Review Board Statement:** Not applicable.

**Informed Consent Statement:** Not applicable.

**Data Availability Statement:** The data that support the findings of this study are available on request from the corresponding author upon reasonable request.

**Conflicts of Interest:** Tosti is an investigator for Eli Lilly and Concert and a consultant for DS Laboratories, Almirall, Thirty Madison, Eli Lilly, Bristol Myers Squibb, P&G, Pfizer, Ortho Dermatologics, and Myovant. The remaining authors have no conflict to declare.

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
