# Peer review of "Cannabinoids for the Treatment of Hair, Scalp, and Skin Disorders: A Systematic Review"

_cosmetics, doi:10.3390/cosmetics10050129_

Round 1

Reviewer 1 Report

Interesting review article related to the Cannabinoids. I only have comment related to the abbreviation especially in the abstract. Please explain first. Also, please prepare the paragraph more than 1 sentence.

Author Response

Comments to the Author
Interesting review article related to the Cannabinoids. I only have comment related to the abbreviation especially in the abstract. Please explain first. Also, please prepare the paragraph more than 1 sentence.

Thank you for your kind and helpful comments. We have now updated the abstract to clarify that CBD is an abbreviation for cannabidiol. We have also re-arranged the introduction to ensure that the first paragraph has more information.

Reviewer 2 Report

The article provides an overview of the use of cannabinoids in the treatment of dermatological diseases and systemic sclerosis. The topic of the article is interesting and relevant. For 5 authors, this is a small review article with a small number of references. We need to expand the introduction. The abstract states that 18 articles were found, but nothing is said about them in the introduction.

What is the GRADE approach? The same can be said in the introduction.

Give comparisons of the treatment of the same diseases with already known drugs. What will be more efficient?

Why did you decide not to submit an article to medical journals?

Author Response

The article provides an overview of the use of cannabinoids in the treatment of dermatological diseases and systemic sclerosis. The topic of the article is interesting and relevant. For 5 authors, this is a small review article with a small number of references. We need to expand the introduction. The abstract states that 18 articles were found, but nothing is said about them in the introduction.

Thank you for your feedback. This is indeed a small review article with a relatively small number of references, but this is a limitation of the currently published literature on cannabinoids in the treatment of dermatological conditions.

We agree that our introduction would be improved with the addition of information, particularly in relation to the findings in the 18 articles we identified. We have now added a paragraph in the introduction with this new information.

What is the GRADE approach? The same can be said in the introduction.

The GRADE approach is a systematic approach to categorizing a study’s quality of evidence as: 1) very low, 2) low, 3) moderate, or 4) high. It provides a systematic approach for making clinical practice recommendations and is the most widely used tool for assessing the quality of evidence. Randomized controlled trials (RCTs) start as high quality evidence, whereas small observational studies start as low quality evidence. Five factors can decrease the quality of evidence (risk of bias, inconsistency, indirectness, imprecision, and publication bias), and three factors can increase the quality of evidence (large effect, dose response, and all plausible confounding). We have now included this information as an additional paragraph in our text (we have chosen to include this in the methods section rather than the introduction as it seems more suitable here).

Give comparisons of the treatment of the same diseases with already known drugs. What will be more efficient?

We agree that comparisons between the efficacy of cannabinoids and established medications in the treatment of dermatologic disorders would be helpful, particularly for providers interested in exploring use of these products. However, because data regarding these comparisons in the form of well-designed studies were not readily available in the literature, we were not able to comment on this in our review. We are hopeful and optimistic that comparative studies will be performed in the future to better elucidate the efficacy of cannabinoids as compared to current standards of care.

Why did you decide not to submit an article to medical journals?

While we could have certainly submitted this article to different medical journals, one of our co-authors was specifically invited to submit an article to Cosmetics’ Feature Papers in Cosmetics 2023 Special Issue, and we felt that this paper would contribute a unique perspective to this special issue.

Round 2

Reviewer 2 Report

The authors did a great job, corrected comments and gave detailed answers to the questions raised. The article is relevant and interesting for readers. I recommend it for publication in  «cosmetics»